# Consensify: A Method for Generating Pseudohaploid Genome Sequences from Palaeogenomic Datasets with Reduced Error Rates

**DOI:** 10.3390/genes11010050

**Published:** 2020-01-02

**Authors:** Axel Barlow, Stefanie Hartmann, Javier Gonzalez, Michael Hofreiter, Johanna L. A. Paijmans

**Affiliations:** Institute for Biochemistry and Biology, University of Potsdam, Karl-Liebknecht-Str. 24–25, 14476 Potsdam, Germany; stefanie.hartmann@uni-potsdam.de (S.H.); thinocorus@googlemail.com (J.G.); mhofreit@uni-potsdam.de (M.H.)

**Keywords:** palaeogenomics, ancient DNA, sequencing error, error reduction, D statistics, bioinformatics

## Abstract

A standard practise in palaeogenome analysis is the conversion of mapped short read data into pseudohaploid sequences, frequently by selecting a single high-quality nucleotide at random from the stack of mapped reads. This controls for biases due to differential sequencing coverage, but it does not control for differential rates and types of sequencing error, which are frequently large and variable in datasets obtained from ancient samples. These errors have the potential to distort phylogenetic and population clustering analyses, and to mislead tests of admixture using D statistics. We introduce Consensify, a method for generating pseudohaploid sequences, which controls for biases resulting from differential sequencing coverage while greatly reducing error rates. The error correction is derived directly from the data itself, without the requirement for additional genomic resources or simplifying assumptions such as contemporaneous sampling. For phylogenetic and population clustering analysis, we find that Consensify is less affected by artefacts than methods based on single read sampling. For D statistics, Consensify is more resistant to false positives and appears to be less affected by biases resulting from different laboratory protocols than other frequently used methods. Although Consensify is developed with palaeogenomic data in mind, it is applicable for any low to medium coverage short read datasets. We predict that Consensify will be a useful tool for future studies of palaeogenomes.

## 1. Introduction

The recovery of nuclear genomic data from ancient biological material—i.e., palaeogenomic data—is typically complicated by high levels of contamination, a low abundance of ancient nucleic acids, and the physical properties of the molecules themselves, such as short fragment length and the presence of miscoding and blocking lesions [1,2,3,4]. Therefore, it can be assumed that the per nucleotide expense of data recovery from ancient samples will be considerably greater than for an equivalent freshly collected sample. Disregarding financial costs, there may also be physical limits on data recovery, as sufficient template molecules may simply not be present for high coverage palaeogenome sequencing of some ancient samples. As a result, published palaeogenome datasets have typically been low coverage (e.g., [5,6,7,8,9,10]), although a small number of high coverage datasets have been published [11,12,13].

Low coverage datasets present a particular challenge for data analysis. Standard single nucleotide polymorphism (SNP) calling approaches involving the identification of heterozygous positions are likely to be error prone when applied to low coverage palaeogenome data, although methods have been developed for bypassing these problems to some extent [14]. Sophisticated methods for estimating SNPs also exist, but these are only applicable for specific datasets such as human SNPs (e.g., [15]). An alternative approach is to disregard heterozygous positions and instead aim to sample a single allele from the sampled diploid chromosomes, producing a so-called pseudohaploid sequence. This can be achieved by majority calling (allele presence calling [16]), which additionally takes into account the presence of ancient DNA damage. However, since the accuracy of majority calling depends on sequencing coverage, applying this approach to datasets with very different levels of coverage (such as modern and ancient datasets) could lead to differential error rates, which may mislead downstream analyses.

Despite the availability of these more complex approaches, arguably the most frequently used approach is to sample a single high-quality nucleotide from the read stack for each position of the reference genome ([6]; hereafter referred to as standard pseudohaploidisation). Assuming equal rates of sequencing and mapping errors among samples, this method effectively downsamples all datasets to 1x coverage, thereby aiming at normalising the rate of errors among datasets. However, the assumption of equal error rates may frequently be violated in empirical palaeogenomic datasets, due to variability in fragment length distributions, levels of cytosine deamination, and laboratory artefacts [17].

Differential errors among pseudohaploid sequences have the potential to confound phylogenetic analyses, principal components analysis (PCA), and other methods for population clustering such as multidimensional scaling (including principal coordinates analysis or classical multidimensional scaling). Increased error rates in certain datasets, for example in ancient compared to modern datasets, are likely to manifest as an excess of singleton sites. For phylogenetic analysis, this will result in an increase in the lengths of terminal branches leading to high-error individuals. Although the internal topology of the tree is less likely to be affected, it is feasible that for more complex analyses that involve constraining the tip ages, the affected lineages could be artefactually pushed to more basal positions in the tree. For population clustering analysis, it is feasible that errors could dominate the variability leading to individuals clustering by error rate rather than ancestry. Furthermore, both absolute and relative estimates of diversity or divergence are likely to be confounded if applied to datasets with substantial differences in error rates.

Several methods have been employed to reduce the effect of differential errors on phylogenetic and clustering analyses. A major cause of sequencing errors in palaeogenomic datasets is cytosine deamination, which manifests as C→T substitutions in sequencing reads [1,2,4]. G→A substitutions can additionally be introduced if blunt-end repair is carried out during library preparation by filling in the 5’ overhangs (reviewed in [18]). A frequently used approach to account for this problem is to exclude transition sites. Although this is an effective means of dealing with transition errors resulting from cytosine deamination, extended terminal branches leading to ancient, relative to modern, samples are nevertheless observed in published phylogenetic trees based on transversions only (e.g., [5]). This suggests that additional sources of error do occur in palaeogenomic datasets at appreciable frequencies. Additional sources of error may include polymerase errors introduced during library preparation [19] as well as incorrect mapping of reads—both of which would result in transition- and transversion-based errors. A potentially useful method to account for these problems is to remove all singletons from the dataset prior to analysis (e.g., [20]). This can be effective if clades are reasonably and equally sampled. However, undersampled divergent lineages will experience a removal of private “real” substitutions and consequently exhibit terminal phylogenetic branch shortening artefacts following singleton removal, as well as potentially fail to form distinct populations in population clustering analyses.

Another class of analyses that may be confounded by differential errors in pseudohaploid sequences includes tests of admixture, such as the frequently used D statistic [6,21]. In its original form, the D statistic uses standard pseudohaploid sequences from two closely related individuals (P1, P2), a third individual representing a candidate admixing lineage (P3), and a fourth individual (P4) that represents the outgroup. Although the D statistic provides a powerful test of admixture, it assumes that alleles are sampled without error [21]. Differential error rates between P1 and P2 individuals present a particular problem, however. For example, if P2 is ancient and P1 is modern, increased errors in the ancient dataset will cause a proportion of its derived alleles to be converted to the ancestral allele, causing it to appear increasingly unadmixed relative to P1 [5,8]. This effect is further magnified when using more divergent outgroups, which has been observed in analysis of empirical datasets, where the D statistic can be shifted from significantly positive to significantly negative solely by changing the outgroup taxon [5]. Recently, efforts have been made to apply a statistical correction to these artefacts. The extended D statistic [22], rather than standard pseudohaploidisation, makes use of the complete read stack and can further apply a correction to error rates estimated by comparison to data from a high-quality “error-free” individual. This method assumes that an excess of singletons in the test dataset relative to the error-free individual is attributed to error, and uses this difference to correct the observed allele counts. In theory, this provides a true error correction by normalising error rates to that of the error-free individual. An implicit assumption, however, is that all individuals are sampled contemporaneously. If the test dataset is from an individual that is appreciably older than the error-free individual, then error rates may be underestimated as the ancient lineage has had less time for substitutions to accrue.

In this study, we present Consensify, a method for reducing error rates in pseudohaploid sequences generated from palaeogenomic and other low to medium coverage datasets. The error reduction is derived directly from the data itself and does not require additional resources such as high coverage data from a close relative, nor does it require simplifying assumptions such as a strict molecular clock or contemporaneous sampling. We show that Consensify brings qualitative improvement over standard pseudohaploidisation for phylogenetic and population clustering analyses. For admixture tests, we also demonstrate, using simulated palaeogenomic data, that Consensify is more resistant to false positives than other available methods, and that it is generally more conservative than other methods when applied to real-world empirical examples. Consensify thus represents a useful tool for future studies of palaeogenomes.

## 2. Materials and Methods

### 2.1. The Consensify Method

Consensify is a simple method for generating consensus pseudohaploid sequences from sequencing reads that are mapped to a reference genome. For each position, three nucleotides are extracted from the read stack at random. If two of the reads agree, then that base is retained. If only two reads are present, but they agree, then that base is also retained. If two reads are present but do not agree, then an N is entered for that position. If coverage is <2, or above a maximum depth specified by the user, then an N is entered for that position. An example is shown in Table 1.

Subsampling reads—of which Consensify is one possible approach—can be formulated in a general form as follows. If n*_i_* is the number of reads covering site *i*, *m* is the minimum acceptable depth for base calling, *M* is the maximum acceptable depth for base calling, *k* is the number of reads used for random subsampling, and *f* is the required number of matching bases for base calling, then: if n*_i_* < *m* or n*_i_* > *M*, then no base call is made.else if n*_i_* > *f*, then *k* reads are randomly subsampledif *f* of the subsampled reads agree, then a base is calledelse, no base call is made.else if n*_i_* = *f*, if *f* reads agree, then a base is calledelse,no base call is made.

In the specific formulation of Consensify described above, *m* = 2, *k* = 3, and *f* = 2. We retain this formulation hereafter and refer to it simply as “Consensify”. Other formulations are possible, however. For example, *m* = *k* = *f* = 1 would correspond to standard pseudohaploidisation, and *m* = 3, *k* = 4, and *f* = 3 would correspond to random subsampling of 4 reads, requiring 3 to agree for base calling (75% majority rule).

To explore the statistical properties of Consensify, we considered a simple model of sequencing error assuming equal genomic base composition and assuming that sequencing errors occur with equal probability across all possible nucleotide combinations. This error model is conceptually identical to the JC69 model of nucleotide substitution [23]. In this model, the global error rate can be summarised by a single variable, *P*g. The probability of observing any base as an error, *P*e, is therefore:
*P*e = *P*g/4,
(1)
For any homozygous position, the probability of observing the correct base, *P*hom, in a single sequencing read is:
*P*hom = (1 − *P*g) + *P*e,
(2)
The last term in the equation reflects that, according to the model, it is possible for a sequencing error to replace a base with the identical base.

For heterozygous positions, the probability of observing a correct base, *P*het, is higher, since an error may convert a base to either allele, thus:
*P*het = (1 − *P*g) + 2 *P*e,
(3)

Sampling three nucleotides mapped to a single genomic position has 64 possible outcomes. By applying this model of sequencing error, it is possible to calculate the probability of observing each outcome given a particular genotype. Summing the relevant probabilities allows the probability of observing a correct base, missing data (N), or an incorrect base, using the Consensify method to be calculated (Appendix A). We calculated expected error rates for Consensify assuming this model and compared them with expectations for standard pseudohaploidisation.

We additionally explored the performance of Consensify when multiple alleles are represented at different frequencies in the read stack. Such a situation may arise due to reference bias, when reads from an individual that is heterozygous at a given position are mapped to a reference possessing one of these alleles. Using the simple model, we calculated the expected probability of sampling an allele which is underrepresented in the read stack (minor allele) using Consensify and compared this with expectations for standard pseudohaploidisation.

### 2.2. Test Datasets

We tested the Consensify method using published Illumina paired-end sequencing datasets of several bear species [5,24,25,26,27]. These comprised three brown bears (*Ursus arctos*), two polar bears (*Ursus maritimus*), an Asiatic black bear (*Ursus thibetanus*) and four Late Pleistocene cave bears (*Ursus spelaeus* complex). Cave bears are suited for testing analytical methods for palaeogenomes because they: (1.) are of Pleistocene age and from temperate environments, and as such will exhibit very advanced DNA fragmentation and nucleotide misincorporations; (2.) have data from multiple individuals available; and (3.) have a well characterised instance of admixture with a living species (brown bears) [5]. The relationship of cave bears to the extant bear species is (black,(cave, (polar,brown))). The cave bear datasets represent four taxa defined based on morphology and mitochondrial DNA, and their relationship is (*kudarensis*,(*eremus*,(*spelaeus*,*ingressus*))) [5]. The brown bear datasets represent individuals from Sweden, Slovenia and Italy, respectively, and their relationship is (Sweden, (Slovenia,Italy)). Full details of the datasets analysed are provided in Table 2.

The cave bear datasets are palaeogenomic datasets that feature the typical properties of ancient DNA [5]. The vast majority of sequences for three of the published cave bear datasets were generated from sequencing libraries prepared using a method based on single-stranded DNA [28], whereas the fourth dataset (*ingressus*, GS136_ds) was generated from sequencing libraries prepared using a method based on double-stranded DNA [29,30]. For this study, we additionally prepared a single-stranded library from DNA extracted from the same petrous bone of the *ingressus* cave bear previously sequenced from only double-stranded libraries, using the method outlined in [28] exactly following the procedure described in [31] and sequenced it on an Illumina NextSeq 500 platform returning 75 base pair (bp) dual-indexed single-end reads, following the procedure described in [32]. These datasets allow a direct comparison of the effect of the library preparation method on downstream analyses.

We additionally modified the Italian brown bear dataset in silico to mimic specific properties of ancient DNA. Ancient DNA fragmentation was simulated by trimming reads to either 35 bp or 50 bp in length using skewer [33]. The effect of cytosine deamination was simulated by using the program TAPAS ([34], available from https://github.com/mlell/tapas) to introduce C→T substitutions around the sequence ends with a proportion of 0.3 at the terminal nucleotides decaying exponentially towards the median nucleotide. An increased global error rate was also simulated using TAPAS by specifying a global misincorporation rate (e.g., sequencing error) of 0.1%. In total, we prepared four simulated ancient datasets (Table 2): 50 bp fragment length with cytosine deamination and sequencing error (simulated palaeo 1); 35 bp fragment length with sequencing error (simulated palaeo 2); 35 bp fragment length with cytosine deamination (simulated palaeo 3); and 35 bp fragment length with cytosine deamination and sequencing error (simulated palaeo 4).

Processing of sequence datasets involved trimming adapter sequences and removing reads < 30 bp using CutAdapt [35]. Overlapping paired-end reads were merged using FLASH [36]. Reads were mapped to the reference genome assembly of the giant panda (*Ailuropoda melanoleuca*; [37]), which represents an outgroup to the investigated clade, using bwa aln [38] and samtools [39], with subsequent filtering for map quality (-q 30) and PCR duplicates (rmdup). The giant panda lineage is relatively diverged from the *Ursus* clade (approximately 12–19 million years; [40,41]), necessitating relaxation of the number of allowed mismatches between read and reference in bwa (-n 0.01; [5]). This is preferable to using the more closely related polar bear reference genome assembly [42] since the polar bear represents an ingroup to the investigated clade, which can lead to biased estimates of admixture [43,44]. Furthermore, polar bears are admixed with brown bears [25], which in turn are admixed with cave bears [5]. Using the polar bear as mapping reference would bias mapping towards the polar bear allele, leading to inflated estimates of admixture with this lineage.

Data processing steps were carried out within the BEARCAVE v.1599d89 data analysis and storage environment (available at: https://github.com/nikolasbasler/BEARCAVE), which provides a convenient resource for data processing and the establishment of a common sequencing data repository. The specific BEARCAVE scripts used were: “*trim_merge_DS_PE_standard.sh*” for trimming and merging paired-end data generated from double stranded libraries; “*trim_merge_SS_PE_CL72.sh*” for trimming and merging paired-end data generated from single stranded libraries; “*trim_SE.sh*” for trimming single-end data; “*map_SE_0.01mismatch.sh*” for mapping ancient data (only merged paired-end reads were mapped for ancient datasets, which are effectively single-end); and “*map_modern_PE_0.01mismatch.sh*” for mapping modern paired-end data. All details of software versions and parameters can be obtained from the BEARCAVE v.1599d89 distribution, which can also be used to replicate the described analyses.

### 2.3. Generation of the Consensify Sequences

To generate a Consensify sequence for each dataset, bases were counted at each position of the reference genome using the -doCounts function in angsd v.0.920 [45], filtered for minimum base quality of 30 (-minQ) and minimum map quality of 30 (-minMapQ). Base counts were not collected for scaffolds < 1 Mb in length (-rf). A custom perl script was then used to perform the Consensify consensus calling described in Section 2.1. This script outputs the sequence in fasta file format with sequence headers matching those of the reference genome, and it calculates the number of successfully called positions. Regions of exceptionally high coverage may represent repetitive elements with accumulations of incorrectly mapped reads. We therefore additionally implemented an optional user-specified maximum read depth filter which can be entered as an integer number. For the purpose of this study, we first calculated the 95th percentile of coverage using the -doDepth function in angsd v.0.920 and implemented the integer number below this value as the maximum allowed depth for consensus calling. The number of Consensify sites successfully called for each dataset is reported in Table 2. The Consensify script is freely available on GitHub (http://github.com/jlapaijmans/Consensify).

### 2.4. Effect of Consensify on Phylogenetic and Clustering Analysis

We compared the performance of Consensify and standard pseudohaploidisation on phylogenetic and population clustering analyses based on genetic distances. These analyses included all modern datasets, all cave bear datasets, and the simulated palaeo 4 Italian brown bear dataset. Genetic distance matrices were computed by standard pseudohaploidisation in angsd, filtered for minimum base quality of 30 (-minQ) and minimum map quality of 30 (-minMapQ), excluding scaffolds < 1Mb (-rf), and only considering sites with zero missing data (-minInd N) that were below the 95th percentile of global coverage (-setMaxDepth), which was determined in advance using angsd (-doDepth). Three distance matrices were calculated by standard pseudohaploidisation including: (1.) all sites; (2.) transversions only (-rmTrans); and (3.) transversions only with singleton removal (1/N < -minFreq < 2/N). A distance matrix was then calculated from the Consensify sequences by combining them into a multi-sequence fasta alignment excluding all columns with missing data, using a custom bash script (‘ReDuCToR’, available from GitHub: http://github.com/jlapaijmans/Consensify). The distance matrix was calculated under the JC69 substitution model using the dist.dna function in the R package *ape* [46,47], considering all sites (both transitions and transversions). Neighbour-joining trees for the four approaches were then calculated using the nj function in *ape* and rooted using the Asiatic black bear as outgroup. For population clustering analysis, distance matrices were re-calculated excluding the Asiatic black bear and principal coordinates analysis carried out using the pcoa function in *ape*.

### 2.5. Effect of Consensify on Admixture Tests

We investigated the performance of Consensify for admixture analysis using the D statistic and compared it with both the D statistic calculated using standard pseudohaploidisation (standard D statistic) and the extended D statistic with error correction applied to the ancient datasets. We calculated D statistics for the phylogeny (((P1, P2), P3), P4). For biallelic sites, alleles sampled in the outgroup are assumed to be ancestral (A) and the alternate allele is therefore derived (B). The D statistic is the difference in the frequencies of sites where P2 and P3 share a derived allele not found in P1 (so called ABBA sites) and those where P1 and P3 share a derived allele not found in P2 (so called BABA sites), normalised for the number of observations. D scales between −1 and +1, with positive values (excess of ABBA sites) suggesting admixture between P2 and P3 subsequent to the divergence of P1 and P2, and negative values (excess of BABA sites) suggesting admixture between P1 and P3 subsequent to the divergence of P1 and P2. Significance of the observed D values was assessed using a 5 Mb weighted block jackknife test with Z-scores > 3 being considered as statistically significant. D statistics were calculated from the Consensify sequences using the published C++ program D_stat.cpp, and the results were processed using the python scripts D-stat_parser.py and weighted_block_jackknife.py ([5], available from https://github.com/jacahill/Admixture). Standard D statistics were calculated in angsd (-doAbbababa1) excluding transition sites (-rmTrans). Sites were further filtered for minimum base quality of 30 (-minQ) and minimum map quality of 30 (-minMapQ), excluding scaffolds < 1MB (-rf), and only considering sites that were below the 95th percentile of global coverage (-setMaxDepth). The standard D statistic results were processed using the R script jackKnife.R, which is included in the angsd distribution. Extended D statistics were also calculated in angsd (-doAbbaBaba2) using the same filters. Error rates in the ancient datasets were estimated using the high-quality modern Asiatic black bear dataset as the error-free individual and the giant panda genome sequence as outgroup. A majority rule consensus fasta sequence was generated from the Asiatic black bear bam file with map and base quality filters (30) using angsd (-doFasta 2) prior to error estimation. Error rates were then estimated for each ancient sample relative to this high-quality consensus sequence in angsd (-doAncError), considering only scaffolds > 1 Mb with map and base quality (30) filters applied. The error correction was applied to the extended D statistic ABBA and BABA counts using the R script estAvgError.R, which is included in the angsd distribution.

We first compared the performance of the three D statistic methods using the four simulated palaeogenomic datasets (simulated palaeo 1–4). Among the three sampled modern brown bears, the Slovenian and Italian individuals are more closely related to each other than either is to the Swedish individual ((Slovenia, Italy), Sweden). D statistic analyses find no evidence that the Slovenian and Italian populations are differentially admixed with the Swedish population (Z < 3; see results Section 3.3) and thus provides a suitable null model. We then repeated these D statistic analyses, substituting the modern Italian brown bear dataset with each simulated palaeogenomic dataset, using both the polar bear (SRS412584) and the Asiatic black bear as outgroup. Since the individuals selected for these tests show no evidence of admixture, any significant D value can be interpreted as a false positive resulting from the modifications made to the simulated palaeogenomic datasets.

We then assessed the effect of library preparation methods on D statistics by using the double- and single-stranded *ingressus* cave bear datasets as P1 and P2, respectively, with all other cave bears as P3 and the Asiatic black bear as outgroup. Since P1 and P2 represent one and the same individual in these tests, the theoretical expectation is D = 0. Any significant deviation from zero can therefore be interpreted as an effect of library preparation confounding the admixture test results. We additionally tested for (1.) admixture among all combinations of cave bears compatible with their species tree (that is, that the test recapitulates their true phylogenetic relationships), (2.) for admixture between all cave bears and the brown bear lineage (represented by the Slovenian individual) subsequent to the divergence of brown bears and polar bears (represented by individual SRS412584), and (3.) for differential brown bear admixture among all cave bear pairs. These tests all used the Asiatic black bear as outgroup.

## 3. Results

### 3.1. Statistical Properties of the Consensify Method

Application of the simple model of sequence error revealed key properties of the Consensify method. For both Consensify and standard pseudohaploidisation, error rates are lower for heterozygous positions than for homozygous positions, but in both cases, Consensify yields substantially lower error rates overall (Figure 1a). For standard pseudohaploidisation, error rates scale linearly with global sequencing error, whereas for Consensify the error rate scales exponentially (Figure 1a). As a result, although the absolute difference in error rates provided by the two methods increases with global sequencing error (Figure 1a), the ratio between them reduces (Figure 1b). For example, under the assumptions of the model, Consensify provides an approximately 130-fold reduction in error rate compared with standard pseudohaploidisation at a global error rate of 1%, and an approximately 27-fold reduction at a global error rate of 5% (Figure 1b). The simple model also predicts an increase in the probability of sampling the most abundant allele with Consensify when two alleles are not equally represented in the read stack (Figure 1c). For example, under the assumptions of the model, the probability of sampling an allele represented with a frequency of 40% in the read stack is 35.2% using Consensify. The probability of sampling alleles using standard pseudohaploidisation is, in contrast, equal to their frequency (Figure 1c).

### 3.2. Effect of Consensify on Phylogenetic and Clustering Analyses

Distance matrices used for neighbour-joining phylogenetic analysis were calculated by standard pseudohaploidisation from 327,736,683, 242,301,611, and 589,004 filtered variable positions for the all sites, transversions only, and transversions only with singleton removal treatments, respectively. The alignment of Consensify sequences included 101,925 filtered variable sites. Phylogenetic analysis generally recovered the expected clades for all treatments, except for standard pseudohaploidisation including all sites (Figure 2a) and all transversion sites (Figure 2b), which failed to recover the expected relationship among the sampled brown bears. The most notable difference observed between treatments were variations in branch lengths (Figure 2). Using all sites resulted in extremely long and variable branch lengths for both cave bears and the simulated ancient brown bear dataset, consistent with increased and differential rates of error (Figure 2a). Filtering for transversions only produced a similar pattern but with less extreme branch lengthening (Figure 2b). Additionally, the double-stranded *ingressus* dataset, which represented the longest terminal cave bear branch when using all sites, is the shortest terminal cave bear branch in the phylogeny calculated from transversions only. Using transversions only with singleton removal produced a phylogeny with more clocklike evolution overall, but with evident branch shortening effects on the more divergent terminal lineages, such as the Swedish and Slovenian brown bear lineages, and the *kudarensis* cave bear lineage (Figure 2c). Analysis of the Consensify sequences produced the phylogeny with the most clocklike evolution overall, with all tips approximately aligned except for the double-stranded ingressus dataset, for which a moderate branch lengthening artefact is evident (Figure 2d).

Distance matrices used for population clustering analysis were calculated by standard pseudohaploidisation from 331,815,528, 246,096,710, and 578,459 filtered variable positions for the all sites, transversions only, and transversions only with singleton removal treatments, respectively. The alignment of Consensify sequences included 88,693 filtered variable sites. Ordination of individual datasets along the first and second principal coordinates revealed substantial differences between treatments (Figure 3). Using standard pseudohaploidisation with all sites resulted in separation of the double-stranded *ingressus* dataset from all other individual datasets along the first principal coordinate. The other cave bears separate from one another and from the brown an polar bear datasets along the second principal coordinate (Figure 3a). The modern polar and brown bear datasets are approximately overlaid, with the simulated palaeogenomic brown bear dataset (simulated palaeo 4) notably diverged from this cluster. Since the overall result departs substantially from the expected pattern of divergence between cave bears, polar bears and brown bears, as well as within cave bears and within the polar and brown bear clade, the overall pattern appears to be driven by excessive error rates in the empirical and simulated palaeogenomic datasets. Filtering for transversions only produced qualitatively similar patterns of divergence, but the separation of the double-stranded *ingressus* dataset is less extreme (Figure 3b). Using standard pseudohaploidisation with transversions only and singleton removal produced three clusters corresponding, respectively, to cave bears, brown bears, and polar bears (Figure 3c). Within the cave bear cluster, the *kudarensis* cave bear is distinct from the other cave bear datasets, which matches with expectations based on phylogeny (Figure 2). However, the empirical and simulated palaeogenomic datasets of the Italian brown bear are notably diverged from the other brown bears, which departs from their expected relationships. This likely reflects the increased representation of private alleles of the Italian brown bear since this individual is represented by two datasets, relative to the Swedish and Slovenian brown bears whose private alleles are excluded by the singleton removal filter. Analysis of the Consensify sequences also produced three clusters corresponding, respectively, to cave bears, brown bears, and polar bears (Figure 3d). However, within the cave bear cluster, the double-stranded *ingressus* dataset is distinct from the single-stranded cave bear datasets. The Italian brown bear datasets are not notably diverged from the other brown bears, however, more closely matching the expected relationships.

### 3.3. Effect of Consensify on Admixture Tests

D statistic tests of admixture among the three sampled modern brown bears using the unmodified Italian brown bear data produced non-significant D values across all three D statistic methods, for both polar bear and Asiatic black bear outgroups (Figure 4). Thus, using the unmodified modern data, we find no evidence that the Slovenian and Italian populations are differentially admixed with the Swedish population. Substitution of the unmodified Italian bear data with the simulated palaeo 1 dataset with 50 bp fragment length, cytosine deamination and sequencing error also produced non-significant D values across all three D statistic methods for the polar bear outgroup (Figure 4a), but with the Asiatic black bear outgroup the standard and extended D statistics both produced false positives whereas the Consensify D-value remained non-significant (Figure 4b). This result does not appear to be driven by a loss of statistical power using the Consensify method, since the number of D statistic informative sites sampled by Consensify is approximately equal to the transversion-only standard and extended D statistic methods. Analysis of simulated palaeo datasets 2–4, with 35 bp fragment lengths, using the standard and extended D statistics produced false positives across all treatments for the polar bear outgroup, but the Consensify D values remained non-significant (Figure 4a). With the Asiatic black bear outgroup, these datasets produced false positives for all methods, but the Consensify D values were generally closer to zero and with lower Z scores than those obtained using either the standard or the extended D statistic (Figure 4b).

Comparisons of the double- and single-stranded *ingressus* cave bear datasets as P1 and P2, respectively, with other single-stranded cave bear datasets as P3, produced false positives (significant D values) using all three methods (Figure 5a). Overall, no method produced obviously lower D values, although fewer D statistic informative sites were sampled and standard errors were larger using Consensify.

Admixture tests among all combinations of cave bears compatible with their species tree using the standard and extended D statistics produced significant non-zero D values for all but one comparison (Figure 5b). The single non-significant value tested for differential admixture with the *kudarensis* lineage among *eremus* and the single-stranded *ingressus* dataset. It is notable, however, that substitution with the double-stranded *ingressus* dataset in this test produced significant positive D values using both the standard and extended D statistics (Figure 5b), and that a general effect of increased D values associated with the double- vs. single-stranded *ingressus* datasets was apparent across all tests. D values calculated using Consensify were non-significant for all comparisons, and closer to zero for all comparisons where the standard and extended D values were significant.

Compatible with previous studies [5], tests of admixture between cave bears and brown bears subsequent to the divergence of polar bears and brown bears were significant using all three methods (Figure 6). Tests for differential brown bear admixture among all cave bear pairs in general supported a gene flow event subsequent to the divergence of *kudarensis* and the European cave bear clade (*ingressus*, *spelaeus*, *eremus*), but with two of these comparisons being non-significant using Consensify and one using the standard D statistic (Figure 6). Both standard and extended D statistics supported an additional gene flow event into the *ingressus* lineage, but only for tests involving the single-stranded *ingressus* dataset. All tests among European cave bears were non-significant using Consensify (Figure 6b).

## 4. Discussion

High error rates are intrinsic to palaeogenomic datasets, since they manifest as a direct result of the physical properties of ancient DNA molecules. Methods for reducing these errors are therefore likely to remain a key aspect of ancient DNA research. Consensify achieves this by normalising coverage bias while leveraging the improved accuracy of calling a consensus from multiple reads, thus combining the key features of standard pseudohaploidisation and majority calling, respectively, in a single method. Similar results could be achieved by downsampling datasets to equal coverage and then performing majority calling. However, this approach does not account for variance in coverage among different genome regions, for example as a result of evolutionary divergence, which is elegantly accommodated by Consensify. For the datasets analysed here, we have shown that Consensify produces fewer analytical artefacts across a range of methods than standard pseudohaploidisation and other approaches.

Compared to standard pseudohaploidisation, Consensify produced phylogenetic branch lengths which fitted closer with molecular clock expectations (Figure 2). Although the removal of transitions and singletons from standard pseudohaploid sequences also produced reasonable trees, the branch reduction artefacts associated with undersampled and divergent lineages 21 may be undesirable. This effect could be mitigated by careful sampling, but this may not be possible in all cases and is a difficult solution to implement a priori. Consensify does not suffer such artefacts and may therefore be better suited for analyses with unbalanced or unknown sampling of clades.

One aspect of the test datasets that Consensify failed to fully mitigate are differential errors among single- and double-stranded datasets. Artificial divergence was obvious using Consensify both with phylogenetic and with population clustering analyses, above that occurring with standard pseudohaploidisation with transition and singleton removal (Figure 2 and Figure 3). It is feasible that removing transitions from the Consensify sequences may improve this result, but such an approach would dramatically reduce the number of recovered sites when sequencing coverage is low. Currently, all individual cave bears with sequenced genomes are represented by datasets generated using single-stranded libraries. Thus, it is possible to analyse their evolutionary relationships using Consensify from highly consistent datasets generated using identical methods (Figure 7). The resulting phylogenetic tree shows very clocklike evolution and no branch shortening artefacts as found with singleton removal (Figure 7a). Population clustering returns three distinct groups corresponding, respectively, to the sampled major bear clades (Figure 7b). Although consistency of laboratory protocols thus provides an effective solution, implementing this solution retrospectively for published ancient datasets generated using varying library preparation as well as DNA extraction methodologies would represent a substantial challenge.

Our results indicate a profound effect of differential error rates on D statistics. Based on the analysis of simulated palaeogenomic data, fragment length seems to be the dominant driver of false positives, having a greater effect on D values than the tested levels of cytosine deamination and global sequencing error (Figure 4). This would suggest that a large proportion of errors in ancient DNA datasets results from short fragments being incorrectly mapped, although further investigation would be required before this hypothesis can be strongly supported. Nonetheless, across all simulated ancient DNA treatments, Consensify was more resistant to false positives than both standard and extended D statistics. One factor in the generally more conservative results using Consensify is an increase in standard error values compared with standard and extended D statistics. Although Consensify often sampled fewer D statistic informative sites, absolute numbers were generally in the tens to hundreds of thousands of sites. Thus, non-significant results would not appear to result from insufficient statistical power. This is further supported by the fact that the Consensify D values are always closer to zero in false positive tests using simulated ancient data than the standard and extended D values (Figure 4). We suspect that the increased standard errors may instead reflect the patchy mapping of reads to the divergent panda reference genome, which will be exacerbated at low coverage when only regions with a read depth above two or three are selected using Consensify. This would lead to greater variance when any single 5 Mb block is removed for weighted block jackknife analysis. If this is the case, mapping to a closely related reference (e.g., ancient human data to the human genome assembly) should not produce such large standard errors, but this is currently untested.

However, one drawback of using a closely related reference is that it may be admixed with and/or represent an ingroup to the investigated clade. In such a situation, bias towards the reference allele will vary among individuals proportional to their relatedness to the reference, which can confound estimates of genetic diversity, population affinity and admixture [43]. In this situation, Consensify would likely exacerbate this effect since it preferentially samples alleles that are overrepresented in the read stack. The use of standard pseudohaploidisation will also introduce such biases, albeit at a lower magnitude. Similarly, mapping of low coverage modern data will be less biased towards the reference allele than equivalent palaeogenomic data, but a bias will still exist. Thus, any results of analyses involving mapping to an ingroup or admixed reference should be evaluated with a degree of caution. If available, one straightforward solution is to map instead to an unadmixed outgroup. Since the reference is then equally related to all individuals, bias towards the reference allele will be normalised among individuals and be less likely to mislead downstream analyses using either standard pseudohaploidisation or Consensify.

Further support for the utility of Consensify is provided by D statistic tests of admixture among cave bears. Of all tests performed, these are the most likely to be affected by differential errors as all ingroup individuals are ancient. In line with this, the standard and extended D statistics returned significant values for all but one comparison among cave bears (Figure 5b). If correct, many of these inferred gene flow events are difficult to explain in an ecological context. For example, cave bears from the Caucasus Mountains (*kudarensis*) would have had to admix with those in the west of Europe (*spelaeus*) to a greater extent than the geographically more proximate cave bear populations in eastern and central Europe (*ingressus*). The inferred occurrence of admixture between *kudarensis* and *ingressus* also changes depending on whether double- or single-stranded datasets are used. Using Consensify, no such complex interpretations are required as no significant evidence of admixture is found among *kudarensis* and any one of the sampled European cave bear lineages, or among *eremus* and either *spelaeus* or *ingressus*, which is compatible with evidence from mitochondrial DNA [48,49].

It is surprising that in false positive tests on simulated ancient data, the performance of the extended D statistic was not evidently different to that of the standard D statistic (Figure 4). This was unexpected, since the extended D statistic in theory provides a true error correction. Consensify, in contrast, only reduces the absolute error rate, meaning that the relative difference in error rates among samples likely remains similar. Consensify therefore relies on reducing differences in ABBA and BABA counts occurring due to errors substantially below that occurring due to population processes. This effect is evident from comparisons with the double- and single-stranded *ingressus* cave bear datasets as P1 and P2, where D values were similar and significant across all methods (Figure 5a). Since differences in ABBA and BABA counts in these tests are solely driven by differential errors resulting from different methods of library preparation, their ratio (and the resulting D value) remains largely unchanged using Consensify. When applied in tests among different cave bears, however, Consensify seems to better mitigate the effect of mixed methods of library preparation, since the inferred patterns of admixture were unchanged when the double- and single-stranded *ingressus* datasets were substituted (Figure 5b). Overall, our results therefore suggest that, at least for the datasets included in this study, Consensify provides lower false positive rates and generally more conservative estimates of admixture than the extended D statistic.

A limitation of Consensify is that the amount of sequencing required to achieve a certain number of pseudohaploid sites will be higher than when using standard pseudohaploidisation. Thus, at extremely low coverage, Consensify will not be applicable because very few sites are covered by two or three reads. This problem is exacerbated when more than one dataset has low coverage, since the probability that any one site has sufficient coverage across all datasets is even smaller. Consensify does mitigate these issues to some extent by allowing the use of transitions as well as transversions, and at higher levels of coverage this can even lead to an increase in informative sites compared with standard pseudohaploidisation sampling only transversion sites (Figure 4a). Recent discoveries such as the mammalian petrous bone as a source of high purity ancient DNA [50], and improved knowledge of the distribution of contaminant DNA across different bone structures [51,52] mean that achieving levels of genome coverage suitable for Consensify is increasingly possible. For example, each single-stranded ancient dataset analysed here was generated using a relatively modest sequencing effort, approximate to a single lane of sequencing on a current Illumina HiSeq platform [5]. For these samples, this produced 3.7–6.9 Gb of mapped data resulting in 0.9–1.3 Gb of Consensify sequence, providing over 100,000 variable sites for phylogenetic and population clustering analysis even after strict filtering, and generally tens of thousands of D statistic informative sites. Until the cost of sequencing reduces to a point where all ancient samples with sufficient surviving DNA can be sequenced to very high coverage, Consensify should represent a useful tool for the analysis of palaeogenomes.

## Figures and Tables

**Figure 1 genes-11-00050-f001:**
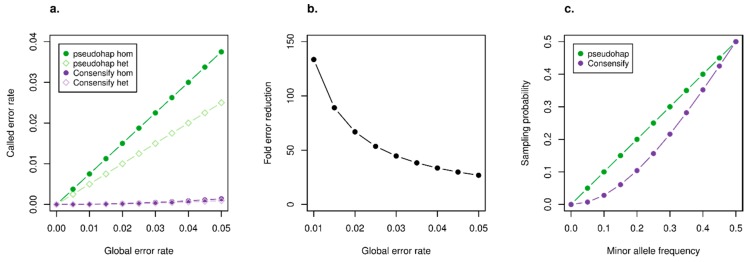
Expected performance of Consensify compared with standard pseudohaploidisation, assuming equal base composition and equal error probabilities across all nucleotides. (**a**) Shows the expected called error rates (y axis) across a range of global error rates (x axis) for standard pseudohaploidisation (green) and Consensify (purple), for both homozygous sites (circles) and heterozygous sites (rhombuses). (**b**) Shows the fold-reduction in error rates achieved by using Consensify compared with pseudohaploidisation (y axis), for a range of global error rates (x axis). Note that the fold-reduction in error is equal for both homozygous and heterozygous sites. (**c**) Shows the probability of sampling (y axis) an allele which is underrepresented in the read stack (x axis) using Consensify and standard pseudohaploidisation.

**Figure 2 genes-11-00050-f002:**
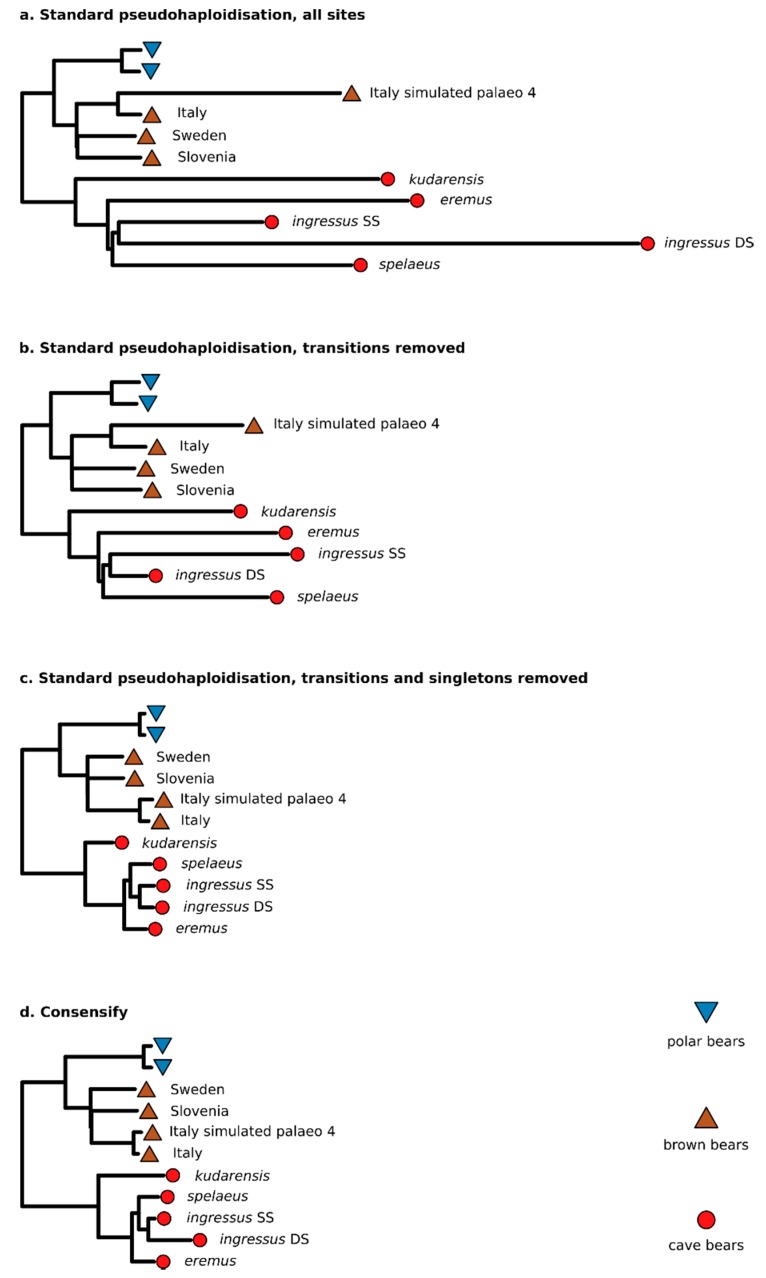
Effect of Consensify on phylogenetic analysis. Panels show neighbour-joining phylogenetic trees calculated from datasets obtained by (**a**) standard pseudohaploidisation using all sites, (**b**) with transitions removed, (**c**) with transitions and singletons removed, and (**d**) Consensify. The trees are rooted using the Asiatic black bear as outgroup (not shown). Coloured symbols at the terminal tips indicate polar bears (blue triangles), brown bears (brown inverted triangles), and cave bears (red circles). The sampling localities of brown bears and the taxon names of cave bears are indicated. “Italy simulated palaeo 4” indicates the simulated palaeogenomic dataset with 35 bp fragment length, cytosine deamination and sequencing error. Note that the *ingressus* cave bear is represented twice, corresponding to datasets generated from sequencing libraries prepared using a single-stranded (SS) and a double-stranded (DS) protocol, respectively. Absolute branch lengths are not comparable among trees because each dataset includes different numbers of sites filtered in different ways. To improve visualisation of relative differences in branch lengths, the trees have been scaled so that the distance between the basal ingroup node and the terminal tips of the polar bear lineage are approximately equal. Polar bears show low genomic diversity [25] and are approaching complete lineage sorting [5], and thus represent the most stable element of the phylogeny with which to anchor the scaling of the trees.

**Figure 3 genes-11-00050-f003:**
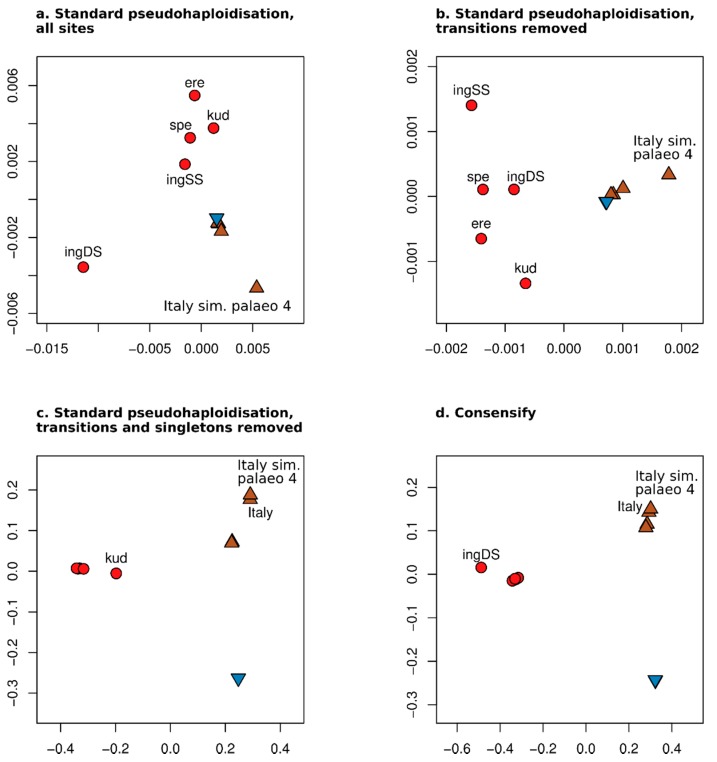
Effect of Consensify on population clustering analysis. Panels show the ordination of individuals along the first (x axes) and second (y axes) coordinates of a principal coordinates analysis based on (**a**) standard pseudohaploidisation using all sites, (**b**) with transitions removed, (**c**) with transitions and singletons removed, and (**d**) Consensify. Coloured symbols are consistent with Figure 1, and, where appropriate, individual cave bears are indicated by the first three letters of their taxon name. “ingDS” and “ingSS” indicate the *ingressus* datasets generated using double- and single-stranded library preparation methods, respectively. “Italy sim. palaeo 4” and “Italy” indicate the simulated palaeogenomic dataset with 35 bp fragment length, cytosine deamination and sequencing error, and the unmodified modern Italian brown bear dataset, respectively.

**Figure 4 genes-11-00050-f004:**
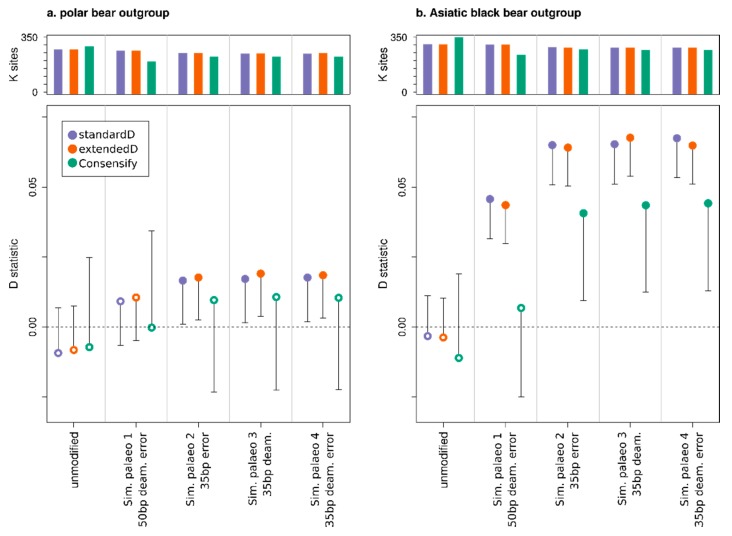
Effect of Consensify on D statistic tests of admixture, evaluated using simulated palaeogenomic data. The tests are based on three brown bears with the relationship: (((P1 = Italy,P2 = Slovenia),P3 = Sweden),P4 = outgroup). Each panel displays results calculated using different outgroups: the closely related polar bear (**a**) and the more distantly related Asiatic black bear (**b**). The upper plot of each panel shows the number of D statistic informative sites (ABBA+BABA, y axes in thousands of sites) counted for each D statistic comparison (separated by grey vertical lines). For each comparison, three results are displayed sequentially from left to right, corresponding to the standard D statistic, the extended D statistic with error correction, and the D statistic calculated using Consensify. The lower plots show D values (y axes) as coloured points. Single error bars extending toward zero show the weighted block jackknife standard error multiplied by three, with error bars that bisect y = 0 (dashed horizontal line) being non-significant (Z < 3). Significant and non-significant D values are further indicated by closed and open points, respectively. The leftmost comparisons in each panel corresponds to the original, high-quality dataset, and does not provide evidence of admixture in any test. For each adjacent comparison, data from the Italian brown bear has been modified in silico to mimic specific properties of palaeogenomic datasets (x axes): short fragment length (35 or 50 bp), C⟶T substitutions increasing exponentially towards the terminal fragment ends (deamination), and increased global sequencing error (error). Any significant D values are therefore false positives resulting from the data modification. Note that y axes are consistent between both panels (**a**,**b**).

**Figure 5 genes-11-00050-f005:**
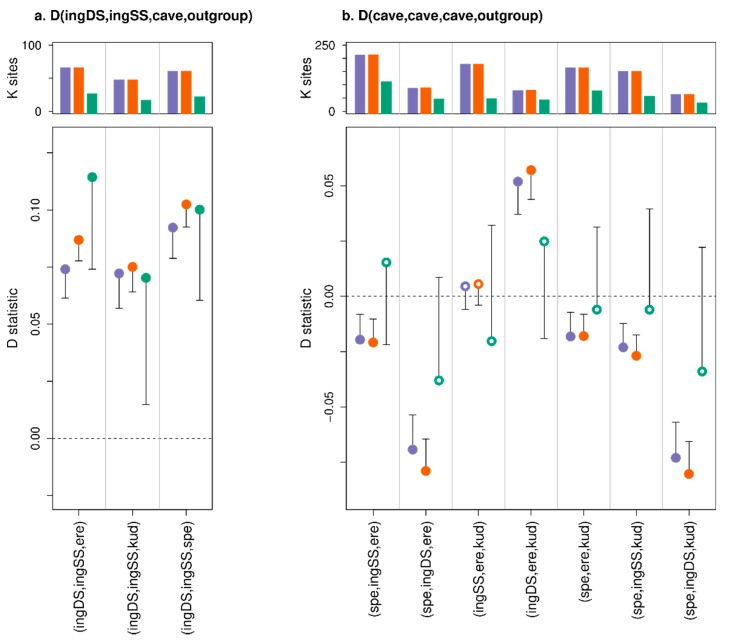
Effect of Consensify on D statistic tests of admixture among cave bear populations and datasets. The plot layout and annotation are consistent with Figure 4. Comparisons are described by x-axis labels, with the first three letters of each cave bear taxon indicating their respective positions as (P1,P2,P3). The outgroup (P4) is the Asiatic black bear. The left panel (**a**) shows comparisons with datasets generated from the same *ingressus* cave bear individual as P1 and P2, corresponding, respectively, to datasets generated using either the single-stranded (SS) or the double-stranded (DS) library protocol. The right panel (**b**) shows all comparisons compatible with the cave bear phylogeny (see Figure 1 and Figure 3): (((*ingressus*,*spelaeus*),*eremus*),*kudarensis*). Note that y axes are not consistent between panels (**a**,**b**).

**Figure 6 genes-11-00050-f006:**
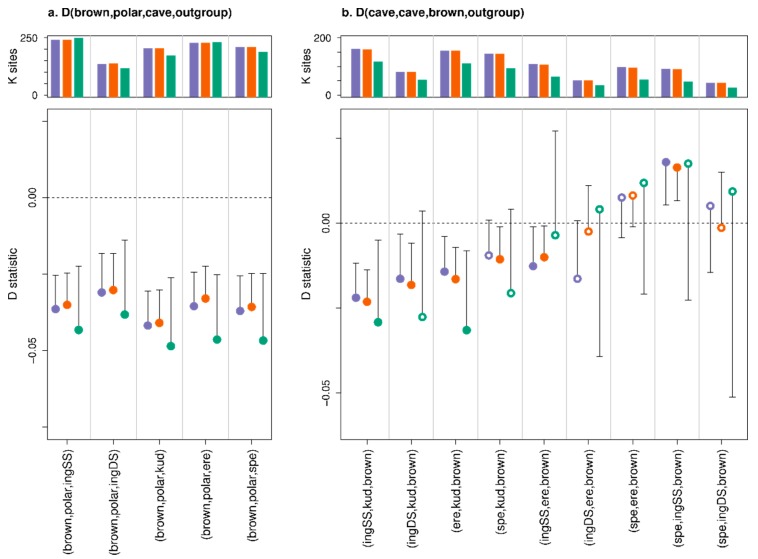
Effect of Consensify on D statistic tests of admixture among cave bears and brown bears subsequent to the divergence of polar bears and brown bears (**a**), and subsequent to the divergence of the sampled cave bear populations (**b**). The plot layout and annotation are consistent with Figure 4 and Figure 5. The polar bear and brown bear lineages are each represented by a single individual (SRS412584 and 191Y Slovenia, respectively). Comparisons are described by x axis labels, with either the first three letters of each cave bear taxon, or “polar” for the polar bear and “brown” for the brown bear, indicating their respective positions as (P1,P2,P3). The outgroup (P4) is the Asiatic black bear. Note that y axes are not consistent between panels (**a**,**b**).

**Figure 7 genes-11-00050-f007:**
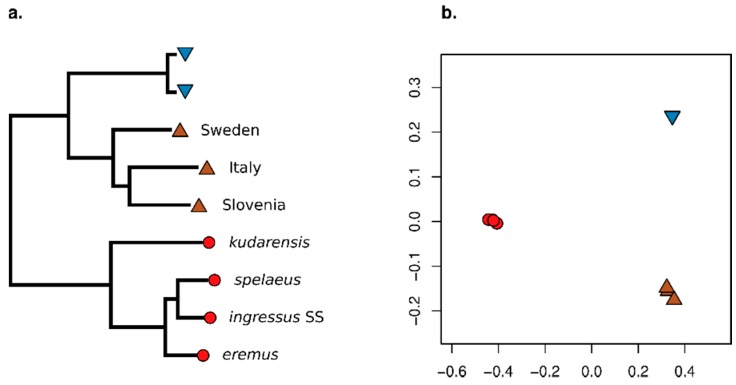
Evolutionary relationships among bears estimated using Consensify. For these analyses, the *ingressus* cave bear dataset generated using the double-stranded library protocol (*ingressus* DS) has been excluded to achieve consistency of methods across all cave bears. (**a**) Maximum-likelihood tree assuming a phylogenetic model of evolution and a GTR+GAMMA model of nucleotide substitution, rooted using an Asiatic black bear as outgroup (not shown). Coloured symbols and tip labels are consistent with Figure 1. (**b**) Ordination of the same individuals along the first (x axis) and second (y axis) coordinates of a principal coordinates analysis.

**Table 1 genes-11-00050-t001:** Summary of a read stack, showing the number of bases observed in columns (totA, totC, totG, and totT) at each position of the reference genome (represented as sequential rows). The Consensify sequence for this read stack would be TGNAC.

totA	totC	totG	totT
0	0	1	2
0	0	2	0
1	0	0	1
4	0	0	0
0	4	1	0

**Table 2 genes-11-00050-t002:** Details of datasets included in this study.

Dataset	Taxon	Data Type	Reference	Mapped Gb ^1^	Coverage ^2^	Prop > 2 Reads ^3^	Consensify Sites ^4^
E-VD-1838	cave bear (*spelaeus*)	ancient single-stranded	[5]	4.55215	1.87465	0.48676	971,153,181
GS136_ds	cave bear (*ingressus*)	ancient double-stranded	[5]	3.72732	1.53497	0.28367	519,642,820
GS136_ss	cave bear (*ingressus*)	ancient single-stranded	this study	6.94074	2.85831	0.64547	1,266,005,835
WK01	cave bear (*eremus*)	ancient single-stranded	[5]	6.12884	2.52396	0.60009	1,210,933,302
HV74	cave bear (*kudarensis*)	ancient single-stranded	[5]	3.76075	1.54874	0.44215	869,048,390
191Y	brown bear (Slovenia)	modern	[5]	6.99668	2.88135	0.54327	1,088,167,390
SRS779830	brown bear (Sweden)	modern	[25]	6.13821	2.52782	0.53406	1,076,186,512
SRR5878360	brown bear (Italy)	modern	[23]	17.5122	7.21182	0.74861	1,553,722,573
		simulated palaeo 1 (50 bp, deamination, error)		10.51792	4.33146	0.74394	1,109,680,441
		simulated palaeo 2 (35 bp, error)		7.55782	3.11244	0.6733	1,297,732,079
		simulated palaeo 3 (35 bp, deamination)		7.42132	3.05622	0.66679	1,293,235,675
		simulated palaeo 4 (35 bp, deamination, error)		7.36212	3.03185	0.66404	1,291,668,999
SRS412584	polar bear	modern	[24]	6.81197	2.80528	0.55527	1,118,483,213
SRS412585	polar bear	modern	[24]	6.02194	2.47994	0.51659	1,025,241,632
ERS781634	Asiatic black bear	modern	[26]	13.43641	5.53334	0.72865	1,523,577,868

^1^ Gb (giga base) data successfully mapped to panda reference genome assembly; ^2^ coverage calculated as total mapped bp divided by the total length of the panda reference genome assembly (2,428,263,693 bp), after filtering for map quality, base quality, and potential PCR duplicates; ^3^ proportion of the genome covered by more than two reads, calculated based only on scaffolds > 1 mega bases (Mb), after filtering for map quality, base quality, and potential PCR duplicates; ^4^ number of sites successfully called using Consensify.

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
