# Peer review of "Consensify: A Method for Generating Pseudohaploid Genome Sequences from Palaeogenomic Datasets with Reduced Error Rates"

_genes, 2020, doi:10.3390/genes11010050_

Round 1
Reviewer 1 Report
The authors introduce Consensify, an improved method for creating pseudo-haploid sequences from low-coverage ancient DNA mapped to a reference genome. Pseudo-haploid sequences, which are a random composite of both parental chromosome haplotypes, can be used in a variety of population genetic analyses, but can be negatively affected by incorrect mapping of short DNA fragments, DNA damage resulting from cytosine deamination, and reference bias. The authors demonstrate shortcomings of the existing method for creating pseudo-haploid sequences (randomly choosing a high quality base at every position), and compare their method to the existing strategies of removing transition and singleton mutations as a way to mitigate the effects of DNA damage.
The authors clearly motivate the problems that their method seeks to solve and demonstrate that Consensify mitigates these problems. They show that Consensify can keep DNA damage from artificially lengthening branches in phylogenies, can prevent artifacts from influencing the major principal components in PCA, and can avoid incorrectly inferring admixture using D-statistics in some cases. They also explore expected error rates in pseudo-haploid sequences produced using Consensify compared to other methods, assuming a standard global error rate.
Broad comments:
Using modern data that they computationally fragmented and/or damaged, the authors infer that the short sizes of ancient DNA fragments posed a greater problem for downstream analyses than DNA damage resulting from cytosine deamination. If interested, another way to test this could be adjusting base qualities in an ancient DNA data set according to the probability that those bases result from DNA damage (using a tool like mapDamage --rescale), then re-generating pseudo-haploid sequences using a base quality score cutoff and seeing whether this improves results.
The authors find that the U. ingressus data set using a double-stranded library preparation technique seems less reliable than that using a single-stranded library preparation technique, but the double-stranded data set is also the lowest coverage ancient sample included in the study. Would the difference in quality disappear if the single-stranded data were downsampled to match the coverage of the double-stranded data?
Although Consensify seems to mitigate problems resulting from ancient DNA damage and fragmentation, the authors also show that it might increase reference bias. Do the authors therefore recommend the existing method for creating pseudo-haploid sequences over their method in the case of low-coverage but high-quality modern data?
Specific comments:
In the last paragraph on page 7, the authors explain a D-statistic calculation that showed no admixture, and they then replace one of the genomes with one that was computationally damaged, repeating the calculation and interpreting any deviation from 0 as a false positive resulting from the simulated damage. It’s not completely clear, though, how the original calculation was done – did the authors use both the polar bear and Asiatic black bear as outgroups, as in the calculations using the simulated ancient DNA damage? And were the data used to infer no admixture produced using all methods under study (standard pseudo-haploidization, standard pseudo-haploidization + transitions removed, and Consensify)?
Figure 1: the authors describe a difference in how the different pseudo-haploidization methods are affected by errors at homozygous and heterozygous sites (e.g. page 4 and Figure 1a). Figure 1b shows the overall fold reduction in error rate using Consensify – is this for both types of sites?
Reviewer 2 Report
This manuscripts describes a method for pseudohaploidization, a process often used in the analysis of low-depth data to compare genomes even if genotyoes can not be reliable called. A particular application for such approaches is ancient DNA data, for which high depth is often too costly or even impossible to obtain.
The proposed method, named consesify, differs from simple random sampling of alleles in that only alleles confirmed by an additional read will be considered. While this puts an additional burden on the required sequencing depth, the authors show convincingly that this extra requirement drops error rates massively, which leads to better estimates of D statistics as well as phylogenetic relationships.
I my view, the detailed comparison of random sampling versus consensify is an important contribution of this manuscript, which is generally very well written and easy to follow. I have therefore only minor comments.
1) Several methods have been proposed for pseudohaplodization, and the manuscript does not do justice to that fact as it presents random allele sampling is the only alternative to consesify. However, many papers use also majority calling, sometimes from downsampled reads, which is very similar to consensify. In addition, Hofmanova et al. 2016 introduced “allele presence calling”, which s now available in ATLAS, in which the most likely allele is identified taking into account sequencing errors and PMD. And there might be additional approaches I’m not aware of. It would be important to mention these other approaches and discuss how consesify differs from those.
In my view, the major difference is that consensify combines using more than a single read with the condition to compare all samples at the same depth. Both majority calling and allele presence calling use all the information provided, and hence do not enforce that the same depth is used for all samples. However, both majority calls and allele presence calls could be filtered for depth or quality, respectively, also effectively ensuring comparable errors across samples.
2) I struggle to see why the authors restrict there approach to a specific depth. Indeed, both majority calling and allele presence calling should out-compete consensify easily at higher depth. Yet, it would be rather trivial to formulate consensify more generally. For instance, one could define consensify as follows.
- Let n_i be the number of reads covering site i
- If n_i < m or n_i > M, no call is made (depth filter).
- If n_i > k, k reads are randomly chosen (downsampling)
- If at least f of the retained reads agree on the same base, that base is called. Else, no call is made.
In the manuscript, the authors chose to use m=2, k=3 and f=2. But other choices could be made. For instance, using m=k=f=1 would correspond to random sampling. Hence I would suggest the authors to describe there method more generally.
3) Please use proper equations when describing the statistical properties, i.e., please use variables rather than English words.
Reviewer 3 Report
The manuscript titled: Consensify: a method for generating pseudohaploid genome sequences palaeogenomic datasets with reduced error rates, is well written. It deals with a very important issue for the field of ancient DNA; how do we make sure that we are using the authentic ancient sequence and not an error incorporated version of it.
By introducing the Consensify method some of this problem gets resolved. The authors explain how this method work in a very concise and convincing way. They do this by using already produced and published data, with the addition of a novel dataset, they go one to show us how it works and how it removes some of the errors. This method can be of great help to the discipline.
The fact that it needs 3 random chosen bases, makes this method hard to use on a lot of the currently generated datasets. Due to the low coverage of a lot of the ancient genomes, having a large set of nucleotides covered three times can be almost impossible. Something that the authors briefly mentions at the very end of the manuscript. This is a weakness of this method. As is the fact that it can’t be used on different types (single, double stranded data) of generated genomes.
This whole method is based upon the 3 randomly selected nucleotides per position in order to generate a consensus pseudohaploid sequence, is that really enough? Should it not take 5 nucleotide readings or more to make sure that you actually have the correct sequence, have the team tested the method for more nucleotides?
Why did they choose to test the method on datasets from bears? Would it not have been better to do this on a domesticate or human dataset?
The use and results of the modified Italian brown bear datasets could be explained better for each subtest. They use it as a “positive control” and as such it is a great idea but they should make the results from these ones easier to find and understand. So that the known, introduced error rate be detected and then it be quantified as a measure of the success rate for this new method. Could they have added a dataset with some known admixture in it to test if this also can be detected using this method?
One minor thing, in Figure 1a it is hard to see the two Consensify lines.
